# Tuberculosis related barriers and facilitators among immigrants in Atlantic Canada: A qualitative study

Isdore Chola Shamputa[1]*, Moira A. Law[2], Clara Kelly[1], Duyen Thi Kim Nguyen[3,4], Tatum Burdo[5], Jabran Umar[5], Kimberley Barker[3], Duncan Webster[6,7,8]

**1** Department of Nursing & Health Sciences, University of New Brunswick, Saint John, New Brunswick, Canada, **2** Department of Psychology, University of New Brunswick, Saint John, New Brunswick, Canada, **3** Government of New Brunswick, Department of Health, Saint John, New Brunswick, Canada, **4** Faculty of Business, University of New Brunswick, Saint John, New Brunswick, Canada, **5** Dalhousie University New Brunswick, MD Program, Saint John, New Brunswick, Canada, **6** Division of Microbiology, Department of Laboratory Medicine, Saint John Regional Hospital, Saint John, New Brunswick, Canada, **7** Dalhousie Medicine New Brunswick, Faculty of Medicine, Dalhousie University, Saint John, New Brunswick, Canada, **8** Division of Infectious Diseases, Department of Medicine, Saint John Regional Hospital, Saint John, New Brunswick, Canada

\* chola.shamputa@unb.ca

**Data Availability Statement:** The data that support the findings of this study are attached to this manuscript as S1 Appendix.

## Abstract

Tuberculosis (TB) is a disease caused by the bacterium *Mycobacterium tuberculosis* and affects approximately one-quarter of the world's population. Immigrant populations in Canada are disproportionately affected by TB. Canada's immigration medical examinations include screening for active TB but not latent TB infection (LTBI). In LTBI, the bacterium remains dormant within the host but can reactivate and cause disease. Once active, TB can be transmitted to close contacts sharing confined spaces leading to the possibility of outbreaks in the broader community. This study aimed to 1) assess the current TB knowledge, perceived risk, and risk behaviors of immigrants in Atlantic Canada as well as 2) identify barriers and facilitators to testing and treatment of TB among this population. Three focus group discussions were conducted with a total of 14 non-Canadian born residents of New Brunswick aged 19 years and older. Data were analyzed using inductive thematic analysis. Four themes were identified from the data relating to barriers to testing and treatment of LTBI: 1) Need for education, 2) stigma, 3) fear of testing, treatment, and healthcare system, and 4) complacency. Results included reasons individuals would not receive TB testing, treatment, or seek help, as well as facilitators to testing and treatment. These findings may inform the implementation of an LTBI screening program in Atlantic Canada and more broadly across the country.

## Introduction

Significant health and healthcare disparities among immigrant populations are a long-standing and well-documented phenomena [1–4]. These disparities adversely impact immigrants'

**Funding:** This study was funded by the University of New Brunswick Research Fund #NF-EXP-2020-07 (I.C.S). The funders had no role in the study design, data collection and analysis, decision to publish, or preparation of the manuscript.

**Competing interests:** I have read the journal's policy and the authors of this manuscript have the following competing interests: Isdore Chola Shamputa serves as an Academic Editor for this journal (PLOS Global Public Health). All the other authors have declared that no competing interests exist.

short-term and long-term well-being [5] and are in no small part due to unmet social determinants of health, such as quality housing, food security, and access to timely and appropriate health care services [6]. The deleterious effects of living in chronic sub-satisfactory life states that immigrants regularly endure are particularly obvious when considering the infection, transmission, and treatment of tuberculosis (TB)—a medical condition typically predicated on social conditions of poverty, pollutants, crowding, and malnutrition [7–9].

## Active and latent forms of tuberculosis

TB is a disease caused by the bacterium *Mycobacterium tuberculosis* and most commonly affects the lungs. There are two forms of the disease, latent and active TB. In latent TB infection (LTBI), the bacterium is dormant within the host and does not cause illness or symptoms of the disease. LTBI screening can be performed with an interferon-gamma release assay blood test (IGRA) or a tuberculin skin test (TST). Active TB causes disease with symptoms that include hemoptysis, night sweats, cough, weakness, chest pain, fever, and weight loss. While active TB is sometimes diagnosed based on the overall clinical picture, the diagnosis generally requires a positive microbiological culture from a clinical specimen. The active form of TB is associated with significant morbidity and mortality if not treated and requires an extensive and lengthy antibiotic regimen. Additional active cases can be prevented and treated through the screening of individuals who are known to have been exposed or identified as being at high-risk, thus preventing negative impacts on the individual and community, e.g., lost time at work and productivity, stigma and discrimination, healthcare-associated costs, morbidity and mortality.

## Screening for active TB and LTBI

The World Health Organization (WHO) estimates that one in four people are infected with *M. tuberculosis* and the disease is estimated to have killed about 1.6 million people in 2021 worldwide [10]. This staggering number is unknown to most within developed countries, where TB is not a major burden on their healthcare system [11–13]. The majority of active TB cases in low TB incidence countries are foreign born individuals [14]. Historically, high-income low-incidence countries have managed migrant TB risks by focusing on early identification of active cases, treatment, and contact tracing for close contacts [15–17]. Conversely, the alternate strategy of selectively screening immigrants from high-burden TB countries for LTBI and offering treatment has been identified as being problematic due to implementation pitfalls and unfinished treatment regimens [18]. However, a recent meta-analysis examining the effectiveness of LTBI screening and treatment programs reported an improvement in completion of treatment regimens for LTBI [19]. For example, a voluntary LTBI screening program for foreign born individuals in Sweden has shown low attrition rates in treatment and an overall cost effectiveness in curbing the costs of potential active TB cases [20]. Other studies have also provided the cost-benefit analysis in favour of LTBI screenings and treatment [21–23], however, many countries continue to only screen for active TB [24] and details on who, when and where to screen remain uncertain [21, 25, 26].

## Screening immigrants to Canada

Currently, Canada has one of the lowest active TB rates in the world [10]. In 2020, Canada reported 1,772 active TB cases of whom 1,303 (75.6%) were foreign born individuals. However, it is estimated over 1.5 million LTBI infections affect the Canadian population [27, 28]. Before entry into Canada, Immigration, Refugee, and Citizenship Canada (IRCC) requires immigrants to complete a chest X-ray to screen for active TB. Should the X-ray reveal signs

consistent with or suggestive of active TB, the newcomer is required to be treated and recovered before entry into Canada [29]. IRCC does not screen or require treatment for LTBI, meaning individuals can enter Canada with the dormant bacteria in their bodies [30]. This leaves immigrants vulnerable to developing active TB and could pose a public health concern if the active disease is not contained. In 2017, Asadi and colleagues compared referral cases, i.e., history of active TB or radiographic indicators of recovery, and non-referrals, i.e., no indication of current or past infection, in the Alberta Tuberculosis Registry, between January 1, 2002, and December 31, 2013, to determine the differential odds of transmission among referrals and non-referrals. They found that referrals were 80% less likely to transmit TB, underscoring the importance of identifying latent TB cases that were responsible for most of the transmission [7].

## The necessity for LTBI testing

New Brunswick plans to welcome 7500 immigrants to the province annually by 2024 [31]. Currently, there is no routine LTBI screening in New Brunswick and immigrants are only screened if they are flagged for medical surveillance based on chest x-ray results or other notable signs. Such is the case in all provinces and territories of Canada [11]. As the number of immigrants continues to grow, screening for LTBI needs to become a priority. Further to infection, immigration increases the risk of TB reactivation within the first 5 years of settlement into a new location [32]. The lack of routine screening for LTBI puts the province and country at risk of having a significant number of active TB cases in a short period of time. Approximately 5–15% of individuals with LTBI will develop active TB [10]. If at-risk individuals are not screened for LTBI and effectively treated, active TB cases are predicted to increase within the newcomer population with the potential for spread to others within the local community [33–35]. Furthermore, this could also promote discriminatory attitudes and behaviors targeted at these immigrants if they are perceived as the instigators of the outbreak, similar to the scapegoating that was witnessed in the early days of the COVID-19 pandemic [36]. The well documented negative impacts of TB related prejudicial attitudes in both high burden TB countries [37–40], and low incidence countries [41–46] consistently reveal the relationship between stigma and reticence in TB testing, delays in initiating treatment, low adherence to treatment regimes, and individuals with long lasting internalized self-stigma [47, 48]. This increase in prejudicial attitudes within the community could in turn worsen long-term health outcomes for the entire community, not just the targeted individuals, due to this increase in prejudicial sentiments toward immigrants [49].

## Barriers to healthcare

When seeking primary care in Canada, immigrants must overcome a series of socioemotional cognitive barriers [50–53], such as poor uptake of relevant health information [54], culturally incompetent clinicians, fear, and discrimination [6, 55]. Barriers may vary in type, e.g., language, and importance, i.e., minor versus major impediments [25, 29, 56, 57]. Depending on the newcomer's unique characteristics, e.g., immigration status, language skills, and social networks, these barriers may be insurmountable when seeking services [58, 59]. These barriers have been categorized as either systemic, e.g., long wait time, or person-specific, e.g., childcare [60]. In Canada, financial burden is not typically identified as a barrier as refugee immigrants are immediately covered by public health insurance under the *Interim Federal Health Program (IFHP)* that covers a wide variety of health care services such as immigration medical exams, treatments, vaccinations, including TB testing and treatment. In the case of economic immigrants there is a three month waiting period from the time of their arrival for their public

health insurance, i.e., Medicare, to be activated bestowing the same benefits as IFHP [61]. Barriers most often cited are cultural issues, language/communication, health system structure, transportation/geographic access, insurance coverage/costs of services, social networks/support, patient-provider relationship, health literacy/immigrants' knowledge, and prior negative experiences that discourage further use of services [49, 50, 57, 58].

Conversely, facilitators are factors that actively support immigrants seeking services such as clinicians taking time to communicate, ask questions, and establish rapport. It also encompasses culturally competent services that consider the individual characteristics of the patient, such as the client's age, gender, and culture [62]. Other facilitators for accessing health care services include avoiding stigma, wanting to protect loved ones from infection, having the right to stay in the country, and wanting to know one's health status [63]. Person-centered delivery of services and health policies that treat each immigrant as a person rather than a population optimize the care of migrant patients [64]. Ensuring financial resources such as health care coverage are consistently cited as important predictors of approaching health care services [65, 66] and adhering to treatment [67].

### Barriers to infectious disease testing

Screening and testing for infectious diseases among immigrants can be met with unique barriers and facilitators [68]. In a recent scoping review, Shehata and colleagues identified low-risk perception of disease, fear of stigma, and fear of testing positive as barriers to testing. Facilitators, from the same scoping review, identified accurate knowledge concerning the contraction/transmission of disease, and having testing as part of routine care [69]. A study examining Chinese immigrants' screening uptake for infectious diseases in Toronto again identified stigma associated with lower rates of screening and facilitators were having a family physician and greater knowledge of the disease [70]. Presently, there is a complete dearth of information regarding barriers and facilitators for immigrants accessing TB screening services in Atlantic Canada.

This study aimed to mitigate current inequities in local public health services to immigrants by assessing their current knowledge of TB, as well as identifying barriers and facilitators to screening and treatment of TB among immigrants in New Brunswick, Canada.

## Methods

### Study setting and participants

This study was composed of a subset of participants (n = 14) who took part in a larger study (N = 43) that investigated the knowledge, attitudes, and beliefs regarding LTBI among documented immigrants in New Brunswick, Canada [71]. To optimize insight into the questions asked, a purposive sampling teachnique was employed, with participants being invited from initial individual interviews via email because they satisfied at least one of the following conditions, they either 1) came from countries with a high TB burden, or 2) during the initial semi-structured interviews they demonstrated relatively high knowledge of TB compared to others in the larger study, or 3) they had experience with an active TB case in their community or family, and were thus likely to provide rich insights [72].

### Study design and data collection

This study used an explorative qualitative research design using focus group discussions (FGDs). The methodological orientation that guided data collection was a phenomenological perspective seeking to understand the participants' lived experience and knowledge of LTBI

testing and treatment. To ensure reporting quality of our study, we relied upon the Consolidated Criteria for Reporting Qualitative Research (COREQ) [73]. Data were collected in three FGDs, which took place between October 7 and November 24, 2020. Each FGD had four to five participants with a combination of male and female participants from different ethnic and cultural backgrounds. All participants had adequate English language conversational skills, hence, no translators were used; notwithstanding, a translator was available for all focus group discussions. The first FGD was held in person, while the other two were held virtually via Microsoft Teams in accordance with social mitigation requirements during the COVID-19 pandemic. Focus groups were selected because they provide a socially interactive environment that likely increases cohesiveness among participants, which can allow them to speak more openly and generate new ideas through verbal interactions [74, 75]. Focus groups were facilitated by two research assistants, with a senior member of the research team present in case there were questions they could not answer. This ensured maximal information, optimal benefits, and opportunities for empowering participants. The FGDs were guided by five main questions informed by prior literature [76–78]; 1) Tell me about your experience with TB in your home country.; 2) Why would some people not want to get screened for TB?; 3) Why would some people who have TB not want to tell their friends or neighbours?; 4) What would make someone with LTBI want to take treatment?; and 5) Is there anything else that you would like to share about LTBI? The progression of questions beginning with their home country was purposeful as it was intended to serve as an easier entry, i.e., familiar, an ice breaker, into what could be a difficult topic.

The FGD guide was piloted for clarity, sequencing of questions, and cultural acceptability with five immigrants who varied by country of birth, sex, culture, and ethnic backgrounds. The FGDs were conducted by two trained research assistants, with one serving as the moderator and the other as a note-taker to take note of verbal and non-verbal cues. At least one co-investigator attended all FGDs. The moderator assigned unique code numbers to each participant to ensure anonymity. Participants stated their assigned code number each time they contributed to the discussion to assist with identifying speakers during the transcription process. All FGDs were conducted in English and lasted between 60 and 75 minutes. No repeat interviews were conducted. Demographic data collected for each participant included country of origin, age, gender, highest education obtained, year of arrival in Canada, employment, whether they had a primary care provider in Canada, and whether they had received the Bacille Calmette-Guérin (BCG) vaccination.

## Ethics approval and consent to participate

Ethical approval was obtained from the University of New Brunswick Research Ethics Board (REB) (file # 034–2020) and the Horizon Health Network REB (file #: RS 2020–2911). Everyone who responded to the larger study recruitment advertisement and showed interest in participating received a copy of the study information document and a consent form via email to assist in their decision of whether to participate in the study or not. Participants were informed that their participation was voluntary and that they could opt out of the study at any time and without consequences. Participants received a $15 e-gift card in appreciation for their time spent participating in the focus group. Confidentiality was assured by using unique codes instead of names and by de-identifying transcripts. All transcriptionists signed confidentiality agreements, and audio recordings and transcripts were saved on a password-protected computer.

## Data analysis

All the FGDs were audio-recorded using two digital voice recorders and transcribed verbatim, except for identifying information, which was removed during the transcription process.

Transcription was conducted by the two research assistants who facilitated the FGDs. The interview transcripts were independently reviewed and verified for accuracy by a co-investigator who attended all the FGDs. Transcripts were not returned to participants for comments and corrections.

Data were manually analyzed using inductive thematic analysis [79]. The method involves six stages: familiarization with the data by reviewing all transcripts, highlighting core concepts in similar colours and generating initial codes. Codes were then reviewed for similarities and groupings of highly related initial codes were merged together. Coders initially worked independently when identifying preliminary codes and conferred when reviewing early emerging themes. Initial themes were shared and discussed with a third coder, culminating with more clearly defined themes that were named and produced the final thematic report.

### Rigor

This study's rigor was ensured by adhering to a previously described framework [80]. Several strategies were employed, including sampling adequacy, which refers to obtaining an appropriate sample for the research topic.

Each FGD was audio-recorded for an accurate analysis, and all discussions were followed by peer debriefing. All interviews were transcribed verbatim. To enhance the rigor of this study, the transcript of one FGD was randomly reviewed against the audio recording to ensure its accuracy. Two co-investigators analyzed the transcripts independently. The team then compared and discussed findings until consensus on themes was achieved. Lastly, an audit trail was maintained to ensure all analysis steps could be traced back to the original data.

## Results

"*I would never go get tested or educate myself about TB because I've never experienced or got personally in contact with somebody who actually had the disease. . .it is after speaking to you. . . I knew that I could be having sleeping latent TB. . .that made me think that I should also go get the test. . .I was never informed about this thing.*"

-Newcomer Focus Group Participant,

Saint John, New Brunswick, Canada, November 2020

### Participant characteristics

Fourteen participants (males = 5 and females = 9) with a mean age of 38.3 (SD = 4.0) years participated in this study. The participants came from eight countries spanning three continents and arrived in Canada within the previous five years. Most of the participants had at least a college education, with about half possessing Master's degrees. This high level of education is characteristic of the broader Canadian immigrant population that account for over half of the graduate degrees in the country [81]. All participants were strangers to each other, except one married couple. Other characteristics of the participants are presented in Table 1.

### Themes

Four major themes emerged from the focus group data that was collected. The themes related to barriers to testing and treatment: 1) need for education, 2) stigma, 3) fear of testing, treatment, and the healthcare system, and 4) complacency. Participants were identified by the FGD

**Table 1. Descriptive statistics of the participants (N = 14).**

| Characteristics | | Male | Female | Total |
|---|---|---|---|---|
| | | N | N | N |
| Gender: | | 5 | 9 | 14 |
| Age: | 25–29 | 1 | 0 | 1 |
| | 30–34 | 0 | 1 | 1 |
| | 35–39 | 1 | 5 | 6 |
| | 40–44 | 3 | 3 | 6 |
| Year of arrival in Canada: | 2006–2010 | 0 | 2 | 2 |
| | 2011–2015 | 1 | 1 | 2 |
| | 2016–2020 | 4 | 6 | 10 |
| Educational level: | College | 2 | 1 | 3 |
| | Bachelor's degree | 1 | 4 | 5 |
| | Master's degree | 3 | 3 | 6 |
| Region of origin: | Africa | 2 | 6 | 8 |
| | South America | 1 | 1 | 2 |
| | Asia | 2 | 2 | 4 |
| Employment status: | Employed | 4 | 7 | 11 |
| | Unemployed | 1 | 2 | 3 |
| Has an assigned primary health care provider (doctor/nurse practitioner): | Yes | 5 | 5 | 10 |
| | No | 0 | 4 | 4 |
| BCG vaccination: | Yes | 0 | 5 | 5 |
| | No | 0 | 1 | 1 |
| | Unknown | 5 | 3 | 8 |

number followed by their assigned participant number, e.g., P.1.5 indicated focus group one, participant number five. Focus group discussion facilitators were research assistants, both immigrants, who introduced themselves to the participants before starting the focus group, each explaining their own roles in the study. The lead researcher, also a foreign born Canadian, was also present in the event a question was raised that the research assistants would not feel capable of answering. They also assisted in clarification of questions when audio equipment malfunctioned for the lead interviewer in one FGD. A translator was available if needed.

**Need for education.** There was a wide variety of gaps in knowledge identified during the FGDs. There was an extended discussion in all three focus groups involving all participants (n = 14) in varying degrees, however, only four of the participants asked questions (Table 2). Participants expressed a lack of knowledge about the existence of LTBI, the disease process from LTBI to active TB, means of transmission, efficacy of the BCG vaccine, and details regarding TB testing. The first prompt in the first FGD, "Tell me about your experience with TB in your home country," quickly evolved into a question-and-answer period with the interviewers, one of whom holds a Ph.D. with a specialty in TB. A total of 14 questions were asked by four participants during this segment of the focus group, with questions ranging from symptoms of TB to the efficacy of the BCG vaccine. All participants came from high TB-burden countries; however, not all participants had direct, e.g., family/friends, or indirect, e.g., someone in the community, experience with TB. Experience with TB appeared to be unrelated to TB knowledge, as both participants with and without experience asked questions (Table 2).

There appeared to be a general lack of awareness of the existence of LTBI among participants. Many did not know there was a latent and active form of TB and did not know the difference between the two forms of TB. For example, P.1.1 stated, "So until I spoke to you in the interview, I had not met or heard [of LTBI]. . .I was totally unaware there was something like

**Table 2. Knowledge gaps: Questions asked by participants during focus groups discussions.**

| Participant | Experience with TB | Knowledge Gaps | Questions (in order they were recorded) |
|---|---|---|---|
| P.5.1 | Yes, experience with extended family member (uncle) | TB transmission, TB testing (timing) | "How it can spread and if we should be careful… or if we should get tested…yeah…so I want to know how much it is contagious… and if a family member has it, is it possible that there is like cancer or any other disease should family members watch out for it…" |
| P.5.1 | Yes, experience with extended family member (uncle) | TB transmission, TB testing (timing) | "…the people who came into contact with that person [with LTBI] needed to be assessed…on which stage of the disease [should someone be tested]?" |
| P.5.1 | Yes, experience with extended family member (uncle) | TB transmission | " and could it spread or transmitted from…[sharing] tea or juices from the same bowl and it is not going into sanitizer…it is just getting rinse in the same bucket again and again…so could that be one of the reason?" |
| P.5.1 | Yes, experience with extended family member (uncle) | TB symptoms | "…people constantly coughing…but could it be related with TB or…they should have some other symptoms with it..like blood..or?" |
| P.5.1 | Yes, experience with extended family member (uncle) | Vaccine protection (BCG) | I had it [the BCG vaccine], is it possible that we have latent TB in our system [now, over 15 years later]…I mean we had the vaccine so we are prevented?" |
| P.4.1 | Yes, experience with core family member (sister) | TB prevalence | "Is TB a tropical disease?" |
| P.5.1 | Yes, experience with extended family member (uncle) | TB prevalence | "Is there an age group…like do we find it more in younger children or in older kids…or adults…?" |
| P.1.1 | No experience with family or friends | Immunity | "Is there a possibility to get it for a second time or it's one time…?" |
| P.2.1 | Not stated | Immunity | "Can we say someone with good immunity can actually survive with TB spread? Can we say that?" |
| P.4.1 | Yes, experience with core family member (sister) | TB testing (current immigration medicals) | "Just a question…during immigration you do a medical…in Canada…I don't know if they look for latent TB in the medical exam?" |

that, so I wanted to do some reading about it. . . and I was amazed that it is something. . .definitely, we should all get tested for". Participant 3.1 indicated they would also get treated, "so it won't become active TB . . . at later years".

Additionally, when asked why some people would not want to get screened for TB, P.1.2 stated, "they don't know the severity of what they are carrying," indicating a lack of knowledge about the disease process. There was consensus among participants, all from different countries, that people were aware a cough was a symptom of active TB; however, it would often be treated with over-the-counter medications or ignored. There was also a sentiment expressed by P.1.3 that some people will not get screened unless they must ". . . nobody wants to [test] proactively. . .like all human beings need to be pushed to get tested. . .when it is mandated. . .unless you are coughing blood or something like that, I think people might take it lightly." Participants asserted that increasing knowledge about the existence of LTBI would have the immediate effect of motivating people to get tested and treated.

In order to address knowledge gaps, participants suggested being proactive in disseminating accurate TB information ". . .through what the [YMCA] Immigrants Centre does for the immigrants [with other information], whatever way to do it [it needs to be done] more proactively. "[Don't wait for] . . .just when it is detected." (P.1.3). An example of primary caregivers stepping into the role of educators was offered by another participant, "When someone is diagnosed with latent TB, to educate the person a little bit more [than they are already doing. That is not enough]. The doctor did tell me, but more information would have helped in terms of keeping you calm. . .I ended up doing a lot of Google search when I was in the hospital" (P.3.1). Participants volunteered to be part of future community initiatives in local settings stating "having different people from different countries, cultural beliefs, and languages [relay this information to the immigrants]." as well as saying "I want to go back to my country and gather the people and educate them" (P.5.1) and "I would volunteer [to educate them] and I know it would do a lot for the community." (P.3.1). They also raised literacy issues when

considering educational outreach programs stating "The government can put as [many] posters out there, but if you can't read and you don't understand, what do you do?" (P.1.2).

**Stigma.**   In many countries across the world, talking about TB is taboo [82]. Participants stated that when their relatives received a positive TB result, the information was kept very secretive even within the family for fear of being shunned. "Because if you declare, if people know you have TB, everybody around you have a possibility to think, if they move close to you, if they talk to you about it, they might be, they might contract . . . so they don't want to have, they don't want to come close to even offer you help or to offer advice" (P.4.2). Lack of knowledge about the disease also contributes to distancing and stigmatization. For example, P.3.1 stated, "When I had latent TB, I didn't tell too many people about it honestly. . . because I didn't want to think it is infectious like active TB . . . because people don't really know the difference". These experiences from their countries of origin influence immigrant's supportive attitudes to avoid TB testing in Canada. For example, P.4.2 stated that if they did test positive, people would "want to run away, they would rather run away, avoid suppers, all together." Participants indicated that providing education to the public about TB and LTBI would help reduce the stigma, and providing information to immigrants, would help them to feel comfortable with screening for LTBI.

**Fear of testing, treatment and healthcare system.**   Several barriers to testing and treatment were identified during the FGDs. Lack of trust and understanding of Canada's healthcare system, perceptions of personal costs of time and money needed for testing and treatment, fear of treatment side effects, uncertainty if they have the mental resources and social support to cope with a positive test, and trying traditional treatments before visiting the hospital as a last resort, were all reported. There were several participants who did not want to be tested for TB because of the perceived inaccessibility of healthcare. In their countries of origin, the testing was very hard to acquire, and if the individual were to test positive, it would be very expensive to treat. Financial barriers in other countries make healthcare inaccessible for many people; for example, P.3.1 stated, "I would check my bank account before I go to the hospital and get myself treated." Others spoke of routinely self-monitoring and medicating with over-the-counter drugs for minor health issues to save money. Participants indicated they did not want to "rock the boat" and avoided seeking medical treatment altogether. For example, P.4.1 stated, "All my life in [country name], I never looked for trouble in myself. . .or go to the hospital looking for sickness. . .because you have to pay for the tests and [then]. . .if you are diagnosed, you have to pay for the treatment".

In addition, there was a mistrust of the healthcare system as P.3.1 stated, "because the basic trust in the healthcare system of our original countries is missing". These barriers due to lack of trust generated in the participants' home countries continue to prevent individuals from seeking care now that they are in Canada. Individuals do not have sufficient knowledge about the intricacies of the healthcare system in New Brunswick, such as accessing free short term public health insurance for some immigrant categories e.g., refugees, creating another barrier to receiving care. Participant 4.1 mentioned other considerations noting "you have time away from work. . .time away from family. . .there will be a lot of other things to think about. . ." if one engages in screening programs or mental health concerns noting "What is my own mental status. . .what if I [am] diagnosed positive. . .how I will handle it. . .how can I find the help?". Participants suggested increasing engagement with available treatment stating "First awareness [of LTBI and free health care], and if I get tested because it's free, and I have it [LTBI], then I get treated, because the treatment is available, [then] maybe [I'll consider] the doctors, maybe the hospital" P.4.2.

**Complacency.**   Complacency was another barrier identified by several participants when asked about why someone might not want to access treatment. "Complacency, as a human

being, out of sight is out of mind. I don't know about an average human being, once the problem it's out of sight, we tend to push it, to keep it down the road. If it is latent, we say, "Oh! Don't worry." P.1.2. This may warrant future investigation to examine immigrants' attitudes towards testing as they relate to personality factors, locus of control, and coping strategies that may pertain specifically to immigrants seeking/avoiding TB screenings in Canada [83].

**Facilitators to testing and treatment.** When participants were asked what would make someone with a LTBI want to take treatment, they stated if a person was to "experience acceptance", "wanted to get better", and had more accurate information available to them, e.g., LTBI and active TB are treatable, immigrants would be more likely to engage in treatment options offered. They (P.3.1 and P.3.3) also felt if immigrants were reminded that it would help keep the broader community healthy, protect their loved ones, and that LTBI will likely become active under the right conditions, it would improve rates of testing and treatment uptake.

## Discussion

The purpose of this study was to gather information to help inform good practices for clinicians, healthcare management, and researchers working with newcomer populations in Atlantic Canada to reduce health and healthcare inequities related to TB testing and treatment. Our results have indicated that there are several barriers to TB screening and treatment within Atlantic Canada, including a lack of knowledge about TB, perceived financial barriers, perceived limited healthcare access, and stigma. These results are in line with recent studies identifying common barriers in relation to TB, including social stigma, inaccessibility to healthcare services, financial constraints, and lack of knowledge about TB [8, 9, 84, 85]. Several participants alluded to the fact that they did not get tested or treated for TB due to the financial implications associated with it. Participants were unaware that if they tested positive for LTBI in New Brunswick, Canada, treatment would be offered to them free of charge. This is in accord with results presented by Pradipta et al. [8], as participants were found to stop taking their TB treatments to continue working, as they were unaware of the available free services.

Participants also exhibited substantial gaps in knowledge concerning basic facts relating to transmission, testing, and TB treatment. This lack of knowledge was surprising given that these participants were chosen based on their previously demonstrated knowledge of TB in the larger study and their immigration from high TB-burden countries. This demonstrates how limited TB information may be known among the average immigrant population. This is concerning, as limited knowledge about TB ultimately contributes to lower diagnoses, fewer individuals treated, and more stigma about the disease [84]. Data gleaned from FGDs offered very specific information needs for this disease in this region, as participants from our study emphasized a need for additional education about the different stages of the disease, as well as more information about the natural history or disease process. Again, this is consistent with findings citing health literacy as a crucial ingredient in delivering effective health services to newcomer communities [86].

Future endeavors to address gaps in TB health literacy, misinformation regarding health care in Canada, and the related stigma that comes with ignorance necessitates the development of educational interventions tailored to this population based on information extracted from these FGD and current published literature [87, 88]. Regarding possible locations for delivering TB health literacy initiatives, participants suggested more information on LTBI should be made available "...through what the [YMCA] Immigrants Centre does for the immigrants [with other information], whatever way to do it [it needs to be done] more proactively. [Don't wait for] ...just when it is detected." (P.1.3). It was also suggested that when immigrants do eventually come in contact with the medical community for either regular care or results from

TB testing to offer more information than is currently being given, "When someone is diagnosed with latent TB, to educate the person a little bit more [than they are already doing. That is not enough]. The doctor did tell me, but more information would have helped in terms of keeping you calm. . .I ended up doing a lot of Google search when I was in the hospital" (P.3.1). It is important patients receive accurate information from reliable sources, such as health care practitioners, rather than searching the internet for pertinent health information that may not be pertinent or accurate [89, 90].

Currently, there are no educational interventions targeting TB knowledge available for either immigrants or the general public in Atlantic Canada. When questions concerning the development of future educational interventions were asked, several participants suggested "having different people from different countries, cultural beliefs, and languages [relay this information to the immigrants]." Participants were so passionate about this aspect of service delivery they said "I want to go back to my country and gather the people and educate them" (P.5.1) and "I would volunteer [to educate them] and I know it would do a lot for the community" (P.3.1). Based on this feedback, program developers might consider recruiting immigrants who have been through the testing program to help in future initiatives, which is consistent with participatory action models that deliver human services by those with lived experience [91, 92]. Another small but crucial consideration participants were concerned with was the translation of educational materials into various languages. They insisted the literacy skills of the immigrants reading this educational information in their own language needs to be considered, "The government can put as [many] posters out there, but if you can't read and you don't understand, what do you do?" (P.1.2).

Participants also offered their opinion on increasing treatment uptake, stating "First awareness [of LTBI and free health care], and if I get tested because it's free, and I have it [LTBI], then I get treated, because the treatment is available, [then] maybe [I'll consider] the doctors, maybe the hospital" (P.4.2). In other words, ensuring immigrants are given solid LTBI information as soon as possible, making sure they are aware that all treatment is free, and by engaging in early treatment they are avoiding doctors and hospitals if the TB becomes active was considered paramount. Campbell and co-workers [93] identified screening for LTBI prior to immigration to a country with a low incidence of TB was an effective, cost-friendly way to reduce the incidence of TB post-arrival. Likewise, Khan and colleagues have found similar success in offering voluntary screening and treatment programs prior to emigration. They had an 88% treatment completion rate for the 21% who tested positive, before arriving in the United States of America [94, 95]. Hence developing strategies and interventions for high TB burden countries to deliver information to emigrants before leaving their country of origin should also be considered and warrants further exploration.

Stigma was also a major barrier to TB testing and treatment, a result that has been noted across many studies [8, 9, 84]. It should be noted that feedback from participants in the focus groups was based primarily on their experience in their homeland; however, these experiences contributed to their current attitudes that continue to influence their behaviour even now living in Canada. Further, by including their homeland in the questions, the bulk of their experience rather than their limited time here in Canada was included in the conversation; conversely, by focusing only on their Canadian experience valuable information might have been excluded. Family members, friends, community members, and even healthcare providers were found to stigmatize individuals with a diagnosis of TB, which created a fear of the disease [8]. Stigma often stems from a lack of information, which was identified by FGD participants. Education on the existence of LTBI, disease progression, and treatment options is imperative to reducing the stigma associated with TB infections for immigrants to the Atlantic Canadian region. A future study focusing on TB-specific health literacy and knowledge gaps appears warranted, given the number of unsolicited questions participants asked during the FGDs (Table 2). Further, understanding what types of stigmas, e.g., enacted stigma, immigrants to

Atlantic Canada experience [96–98] and the specific locales where this occurs would provide insight into how these factors influences quality of care and offer direction on developing local stigma reduction interventions [99, 100].

The current study underscores the necessity for understanding global issues through a local lens [101]. Canada does not have a national LTBI screening strategy for immigrants; thus, this project addresses both a clinical gap in public health and increases understanding of the stigma, fear, and ignorance that may interfere with immigrants accessing healthcare services in Canada. The need for an education program to increase TB knowledge is clear. What is not clear and warrants future attention is if this need for education exists in the broader community. Further, interventions to help immigrants overcome fear of testing, treatment and the health care system are needed to be developed and delivered for the local population. The influence of structural barriers, e.g., laboratory services, and their relationship with identified socio-cognitive barriers, e.g., stigma, need to be understood in the local context. As immigration around the world continues to increase research focusing on undocumented immigrants will also need to be addressed. The findings of this study pertain directly to Atlantic Canada and with future study and time may also show applicability to the rest of our vast and diverse country. Future research based on these findings need to remain sensitive to local immigration characteristics and trends, so by attending to this emerging evidence a helpful response may be mounted to address the local, and perhaps global, need of the migrant [102].

Based on the current study, the following policy recommendations are suggested:

1. the government should promote more aggressive LTBI screening with certain at-risk populations.

2. the government should develop TB and LTBI educational programs for newcomers and healthcare providers and ensure a viable work force is in place to manage both LTBI and active TB cases.

Finally, as we move into the next phases of the COVID-19 pandemic world, there is a heightened awareness of infectious diseases and the necessity for testing and early interventions [103, 104]. This current zeitgeist would support new initiatives for testing programs. Results from this study and others like it [8, 48, 49, 57] highlight the importance of infectious disease screening for more than just TB. Screening for other infectious diseases could be included to establish a broader infectious disease prevention program for Atlantic Canada.

## Authors personal reflexivity

Our research team has been together for four years, with an interdisciplinary strength drawing from medicine, public health, epidemiology, nursing, psychology, diagnostics, population health, and infectious diseases, ranging from seasoned researchers to undergraduate medical and nursing students. We are all drawn together by our common interest in vulnerable populations including immigrants to our local community. We come with a variety of research backgrounds including quantitative, mix-methods, meta-analyses, systematic reviews, and qualitative approaches. Our diverse team has expertise in various key areas relevant to this study, including research design, community engagement, immigrant health, policy implementation, infectious diseases, health inequities, and equity, diversity, and inclusion, to name a few. Below are key details outlining the capacity and credentials of our interdisciplinary team, and our ability to successfully implement our study: Dr. Isdore Chola Shamputa holds a Ph.D. in Medical Sciences from the Vrije Universiteit Brussel in Belgium and has many years of mycobacteriology research experience across Africa, Europe, Asia, and North America. He has advanced training to post-doctoral level in TB research from the National Institutes of Health in the United States. Dr.

Shamputa is also a registered nurse who has experience caring for patients from various backgrounds, including immigrants. As an Associate Professor in the Department of Nursing & Health Sciences at the University of New Brunswick, he has dedicated time for research and mentoring undergraduate and graduate students. Dr. Shamputa brings a wealth of experience leading various qualitative and quantitative research projects and has expertise in public health, epidemiology, project management, research design, and data analysis. Dr Moira Law holds a Ph.D. in Psychology trained at Carleton University, Ottawa, Ontario, Canada and subsequently working in and with a variety of departments in the federal government. She has conducted several meta-analyses and is intensely interested in identifying best practices through the systematic synthesis of the peer reviewed published literature. She deeply values the voices that speak through qualitative researchers' work. She is motivated to make visible marginalized persons in our prisons, institutions and communities. Dr. Duncan Webster is a Royal College of Canada certified medical microbiologist, and the medical director of the level 3 laboratory at the Saint John Regional Hospital in New Brunswick. He also works in the Division of Infectious Diseases, Department of Medicine, Saint John Regional Hospital where he provides clinical care for infectious diseases, including treating LTBI and active TB patients. In addition, he is an Associate Professor at Dalhousie Medicine New Brunswick. Clara Kelly is a member of the 2023 graduating class of University of New Brunswick's Bachelor of Nursing program. As an undergraduate student, she worked on several qualitative research projects with senior members of our team. Dr. Duyen Thi Kim Nguyen holds a Ph.D. in population and public health and completed a 2-year Canadian Institute of Health Research Health Systems Impact Post-Doctoral Fellowship. Dr. Nguyen brings a wealth of experience leading various qualitative and quantitative research projects, and has expertise in public health, social determinants of health, project management, research design, data analysis, scaling-up health interventions, and equity, diversity, and inclusion. Dr. Tatum Burdo is a graduate from Dalhousie Medicine New Brunswick, and also holds a Master in Public Health. Dr. Jabran Umar is also a graduate from Dalhousie Medicine New Brunswick. Both Tatum and Jabran are currently completing their residency training. Dr. Kimberly Barker is a member of the Royal College Physicians and Surgeons of Canada in Public Health and Preventative Medicine, and an Associate Professor at Dalhousie Medicine New Brunswick supervising several medical students who have an interest in TB in immigrant populations. She also holds a Master in Public Health and is currently the Region Medical Officer of Health responsible for southern New Brunswick. She brings many years of experience as a physician, with Public Health training and has managed TB programs in Canada and internationally. Our team holds varied experiences working in clinical, research, government, and academic settings with different goals on different projects. We recognize our collective expertise and subjectivities come with biases that we hope can still serve well those who are being studied in this project.

## Conclusion

The results of this study indicate that there is a need for increased education regarding TB and LTBI within immigrant populations. The lack of knowledge contributes to the stigma around TB and creates barriers to testing and treatment, as individuals are unaware of LTBI and what is available for treatment. Providing comprehensive education about TB and LTBI to all at-risk immigrants should be a priority for the provincial governments in Atlantic Canada to ensure testing is completed and local outbreaks of TB are averted.

### Limitations

The sampling strategy used in this study was not random and hence amenable to potential bias. Only three focus groups were conducted possibly restricting code saturation [105]. Also,

the small sample size and high education level of participants may have limited the saturation [105] of identified themes as well as the overall generalization of our findings.

## Supporting information

**S1 Appendix. Focus group discussions transcripts.**
(PDF)

## Acknowledgments

The authors wish to thank the participants, translators, and research assistants for their help.

## Author Contributions

**Conceptualization:** Isdore Chola Shamputa, Kimberley Barker, Duncan Webster.

**Data curation:** Isdore Chola Shamputa.

**Formal analysis:** Isdore Chola Shamputa, Moira A. Law, Clara Kelly, Duyen Thi Kim Nguyen, Tatum Burdo, Jabran Umar.

**Funding acquisition:** Isdore Chola Shamputa.

**Investigation:** Isdore Chola Shamputa.

**Methodology:** Isdore Chola Shamputa, Kimberley Barker, Duncan Webster.

**Project administration:** Isdore Chola Shamputa.

**Resources:** Isdore Chola Shamputa.

**Software:** Isdore Chola Shamputa.

**Supervision:** Isdore Chola Shamputa.

**Validation:** Isdore Chola Shamputa, Moira A. Law, Duyen Thi Kim Nguyen.

**Visualization:** Isdore Chola Shamputa, Moira A. Law, Duyen Thi Kim Nguyen.

**Writing – original draft:** Isdore Chola Shamputa, Moira A. Law, Clara Kelly, Duyen Thi Kim Nguyen, Duncan Webster.

**Writing – review & editing:** Isdore Chola Shamputa, Moira A. Law, Clara Kelly, Jabran Umar, Kimberley Barker, Duncan Webster.

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
