## [Decision Letter · Decision Letter 0]

23 Jan 2023

PGPH-D-22-01914

Tuberculosis related barriers and facilitators among immigrants in Atlantic Canada: A qualitative study

Dear Dr. Shamputa,

Thank you for submitting your manuscript to PLOS Global Public Health. After careful consideration, we feel that it has merit but does not fully meet PLOS Global Public Health’s publication criteria as it currently stands. Therefore, we invite you to submit a revised version of the manuscript that addresses the points raised during the review process.

We look forward to receiving your revised manuscript.

Kind regards,

Sanghyuk S Shin

Academic Editor

Journal Requirements:

b. If any authors received a salary from any of your funders, please state which authors and which funders.

2. We ask that a manuscript source file is provided at Revision. Please upload your manuscript file as a .doc, .docx, .rtf or .tex.

3. Your manuscript is missing the following sections: Introduction. Please ensure these are present, and in the correct order, and that any references to subheadings in your main text are correct. An outline of the required sections can be consulted in our submission guidelines here:

https://journals.plos.org/globalpublichealth/s/submission-guidelines#loc-parts-of-a-submission

4.  In the online submission form, you indicated that "The data set used and analysed during the current study are available from the corresponding author on reasonable request". All PLOS journals now require all data underlying the findings described in their manuscript to be freely available to other researchers, either 1. In a public repository, 2. Within the manuscript itself, or 3. Uploaded as supplementary information.

Additional Editor Comments (if provided):

Reviewers' comments:

Reviewer's Responses to Questions

**Comments to the Author**

1. Does this manuscript meet PLOS Global Public Health’s publication criteria? Is the manuscript technically sound, and do the data support the conclusions? The manuscript must describe methodologically and ethically rigorous research with conclusions that are appropriately drawn based on the data presented.

Reviewer #1: Yes

Reviewer #2: Partly

2. Has the statistical analysis been performed appropriately and rigorously?

Reviewer #1: Yes

Reviewer #2: N/A

3. Have the authors made all data underlying the findings in their manuscript fully available (please refer to the Data Availability Statement at the start of the manuscript PDF file)?

Reviewer #1: Yes

Reviewer #2: Yes

4. Is the manuscript presented in an intelligible fashion and written in standard English?

Reviewer #1: Yes

Reviewer #2: Yes

5. Review Comments to the Author

Reviewer #1: The authors have clearly summarized the perceptions of recent immigrants to Canada about TB prevention and care services. Findings are in line with the literature. Please review the COREQ checklist and include missing items in the manuscript's methods section.

Reviewer #2: Thank you for the opportunity to review this manuscript. It adds new information to an important topic. My comments can be found below:

Background

- The background section should provide more context regarding the number of people with TB and LTBI diagnosed each year. Of the ~7,500 immigrants to New Brunswick each year, how many are currently screened for LTBI? The scale of the TB problem is unclear.

- The manuscript should provide more information about the structure of the health system in New Brunswick, health financing model and the cost of accessing TB care. The manuscript does not make it clear in the background whether immigrants would be asked to pay for care out of pocket. Do all of these immigrants have access to health insurance in Canada? What about undocumented immigrants? Adding this information will help the reader understand when participants discuss affordability of care (e.g. Lines 281- 282) when you wrote, "Individuals do not have sufficient knowledge about the complex healthcare system in New Brunswick..." Currently, the reader is left in the dark about the structure of the healthcare system and what makes it complex or challenging to navigate.

- One of the main findings of this paper is that immigrants need more education (Lines 395-396) about TB and LTBI, but it is unclear what education and outreach is provided in the region.

- Lines 116- 119. Facilitators of healthcare also include person-centered policies, accessibility of health financing, etc. Additional literature should be cited here.

- The background is solely focused on Canada, when other high-income countries are also adopting similar strategies to address LTBI. To increase the applicability of the manuscript to other contexts, it would be helpful to include some additional background information on the various LTBI screening strategies and policies being employed in other contexts.

- Lines 96-104. There is a long history of stigma and discrimination for people with TB and LTBI, yet this paragraph focuses on potential scapegoating similar to that seen during the early days of the COVID-19 pandemic. If this scapegoating of immigrant communities happened in previous TB outbreaks in Canada, then a more concrete example should be referenced, rather than speculation about future scapegoating.

Methods

- Lines 140-141. The sampling structure should be clarified. It was unclear what "demonstrated high knowledge of TB in the larger study" means. Was this demonstrated through a survey or interactions with study staff? This is an important clarification because the discussion section ( Lines 327-338) seems to identify that the participants did not in fact have "high knowledge." Did the sampling method produce a group that was able to richly discuss the topic? If the participants were not able to do this, then how can the researchers claim that "sampling adequacy" (Line 193) was obtained.

- Further questions about sampling. Does the high education level of the participants match the overall immigration profile of Canadian immigrants or were higher-than-average educated participants purposively sampled? Were any of the participants related?

- In the FGDs, five questions were asked by the researchers and none specified anything about Canada or New Brunswick. It is unclear why the decision was made to collect information about TB in both home countries and Canada to then draw conclusions only about Canada.

- Line 163. Did conducting the interviews in English cause any challenges?

- Lines 191- 200. The section on rigor should explicitly state the methodological orientation of the study, e.g. grounded theory, discourse analysis, content analysis, etc. This manuscript would generally benefit from reviewing and incorporating the COREQ (COnsolidated criteria for REporting Qualitative research) Checklist.

- Table 1. How many participants had direct experience with TB treatment? The descriptive statistics should include previous history of TB/LTBI treatment, or living in the household of someone with TB, if available.

Results

- Theme 3 is about the barriers and facilitators of LTBI testing. However, this essentially restates your research question. A theme should not be this close to the research question; it should elucidate it. Authors should consider revising this theme entirely.

- Lines 215- 225. This section seems to indicate that FGDs devolved into a question-and-answer session, where the expert never ceded their position of authority to the participants. More information should be provided about the decision to select a TB specialist as the primary interviewer and the information that the participants were provided about the researcher. Lastly, the manuscript should provide a reflexivity statement that acknowledges and discusses the researcher's role in the qualitative interview process.

- Line 238. Remove the contraction, unless it is supposed to be part of the quote.

- Table 2

- At times the theme is written as "TB testing (timing)" and other times "Timing of testing". If this is one theme, then label consistently, or clarify the differences.

- In the results the participants are labeled P dot x dot x (P.3.1) In this table, they are labeled as Px dot x (P4.1). Please use consistently throughout.

- 60% of questions were asked by P5.1. Two more (20%) were asked by P4.1. Did these two participants dominate the Q&A session? Or did all participants ask questions?

- The Stop TB Partnership, the UN-agency dedicated to TB, has issued a TB Stigma Assessment Tool. This tool defines various forms of TB stigma including: anticipated stigma, self-stigma, enacted stigma, secondary TB stigma, perceived TB stigma, structural stigma, etc. This manuscript would benefit from a more nuanced discussion of the concept of TB stigma, which is already in wide circulation among the TB community.

- Lines 263- 267. Please clarify this sentence. It is confusing with multiple uses of e.g.

- Section on Barriers to testing and treatment. Why focus the results on barriers in both home countries and Canada, and then the discussion is limited to best practices in New Brunswick to reduce health inequities? There seems to be inconsistency between the results and findings. Secondly, the barriers focus on "socioemotional cognitive barriers" without addressing any structural barriers in Canada.

Discussion

- Lines 306- 312. This should be included in the results and not the discussion.

- The discussion should better locate the sites of experienced/perceived TB stigma in Canada, in order for them to be directly addressed.

- Lines 352-354. "...culling selected information based on a misinformed confirmation bias." Consider revising sentence to place less blame for a lack of information on people with TB.

- Lines 376-378. It may be beneficial to reference the work done in the U.S. by Amera Khan on screening for LTBI among US-bound immigrants here, along with examples from other high-income, low-burden TB countries.

- The discussion focuses on "strategies and interventions" for LTBI but leaves out immigration policy change. Consider adding policy recommendations.

- The manuscript describes the geographic coverage as "Atlantic Canada" and "New Brunswick" but then in the abstract and conclusion, it is described as relevant for "southern New Brunswick". Please clarify the applicability of the findings. Are the authors making recommendations only for one region of Canada? Could recommendations be made applicable to all of Canada?

Conclusion

- The manuscript calls for more education among immigrants, but leaves out healthcare workers, the general population, etc. Immigrant communities are not the only source of TB stigma, and an effective anti-TB stigma intervention TB stigma will certainly need to address multiple forms/loci of stigma.

6. PLOS authors have the option to publish the peer review history of their article (what does this mean?). If published, this will include your full peer review and any attached files.

**Do you want your identity to be public for this peer review?** For information about this choice, including consent withdrawal, please see our Privacy Policy.

Reviewer #1: No

Reviewer #2: No

---

## [Decision Letter · Decision Letter 1]

5 Apr 2023

PGPH-D-22-01914R1

Tuberculosis related barriers and facilitators among immigrants in Atlantic Canada: A qualitative study

Dear Dr. Shamputa,

Thank you for submitting your manuscript to PLOS Global Public Health. After careful consideration, we feel that it has merit but does not fully meet PLOS Global Public Health’s publication criteria as it currently stands. Therefore, we invite you to submit a revised version of the manuscript that addresses the points raised during the review process.

We look forward to receiving your revised manuscript.

Kind regards,

Sanghyuk S Shin

Academic Editor

Journal Requirements:

Additional Editor Comments (if provided):

Reviewers' comments:

Reviewer's Responses to Questions

**Comments to the Author**

1. If the authors have adequately addressed your comments raised in a previous round of review and you feel that this manuscript is now acceptable for publication, you may indicate that here to bypass the “Comments to the Author” section, enter your conflict of interest statement in the “Confidential to Editor” section, and submit your "Accept" recommendation.

Reviewer #2: All comments have been addressed

2. Does this manuscript meet PLOS Global Public Health’s publication criteria? Is the manuscript technically sound, and do the data support the conclusions? The manuscript must describe methodologically and ethically rigorous research with conclusions that are appropriately drawn based on the data presented.

Reviewer #2: Partly

3. Has the statistical analysis been performed appropriately and rigorously?

Reviewer #2: N/A

4. Have the authors made all data underlying the findings in their manuscript fully available (please refer to the Data Availability Statement at the start of the manuscript PDF file)?

Reviewer #2: No

5. Is the manuscript presented in an intelligible fashion and written in standard English?

Reviewer #2: Yes

6. Review Comments to the Author

Reviewer #2: The authors did significant work to improve the quality and clarity of the manuscript since my first review. However, the manuscript still requires extensive edits.

When I went to check the data availability at the University of New Brunswick, I was unable to find the data when I search for both the manuscript title or the name of the first author. The DOI link is invalid.

Line 94- missing a comma in 1,303.

Lines 174-180- The number of participants recruited through each strategy should be made clear. How this study implemented the strategy of sampling adequacy is not described sufficiently in the methods section.

Table 1- You have described region of origin, not countries so the label on page 12 needs to be updated. Secondly, it is unclear for individuals who are not familiar with the Canadian healthcare system whether these individuals have been linked to, assigned to, or selected a primary healthcare practitioner or whether they are in fact working as healthcare providers themselves.

Line 271-272- This sentence should be in the methods and not the results.

Lines 228-240- You should mention more about the coding of the interview transcripts. How was this done. Was a coding tree created? What software was used to manage the data? This is also in the COREQ checklist and standard for reporting qualitative research.

Line 280- This provides insufficient information about the information that the participants were provided about the interviewers. This is a component of the COREQ checklist and can easily be added to the manuscript.

Table 2- The table lists seven themes that are not discussed as themes in the manuscript. This inconsistency should be clarified. This table also does not represent the entire sample well. Only four individuals asked questions and 6/10 questions come from one individual.

Line 364- I appreciate that the authors have worked to clarify whether an individual was discussing active TB or LTBI throughout the manuscript. However, in this line, it seems odd to mention LTBI and hospitalization. In which context would people be hospitalized for LTBI?

Lines 394-424- You are presenting new data in the discussion. All quotations should be first presented in the results section.

This manuscript would still benefit from a reflexivity statement. In the response comments, the authors state that reflexivity has been addressed through lines 195-198. This is not a reflexivity statement.

COREQ checklist- The authors have insufficiently used these reporting guidelines.

Q2, Q3, Q5. Researchers credentials, education and experience could be included in the manuscript.

Q7. I addressed this in the comment above.

Q18 & Q23.- Why did you sometimes answer N/A vs. No? If you did not include repeat interviews then this can be easily addressed both in the manuscript and the checklist.

Q30. I would say that you did in fact do this. When you illustrate the discussion with the data presented in the results, it can be listed here.

Q32. I would also say that you discuss minor themes in your results section. I do not see how the response of not applicable applies to the discussion of minor themes.

7. PLOS authors have the option to publish the peer review history of their article (what does this mean?). If published, this will include your full peer review and any attached files.

**Do you want your identity to be public for this peer review?** For information about this choice, including consent withdrawal, please see our Privacy Policy.

Reviewer #2: No

---

## [Decision Letter · Decision Letter 2]

11 May 2023

Tuberculosis related barriers and facilitators among immigrants in Atlantic Canada: A qualitative study

PGPH-D-22-01914R2

Dear Dr. Shamputa,

We are pleased to inform you that your manuscript 'Tuberculosis related barriers and facilitators among immigrants in Atlantic Canada: A qualitative study' has been provisionally accepted for publication in PLOS Global Public Health.

Best regards,

Sanghyuk S Shin

Academic Editor

Reviewer Comments (if any, and for reference):

Reviewer's Responses to Questions

**Comments to the Author**

1. If the authors have adequately addressed your comments raised in a previous round of review and you feel that this manuscript is now acceptable for publication, you may indicate that here to bypass the “Comments to the Author” section, enter your conflict of interest statement in the “Confidential to Editor” section, and submit your "Accept" recommendation.

Reviewer #2: All comments have been addressed

2. Does this manuscript meet PLOS Global Public Health’s publication criteria? Is the manuscript technically sound, and do the data support the conclusions? The manuscript must describe methodologically and ethically rigorous research with conclusions that are appropriately drawn based on the data presented.

Reviewer #2: Yes

3. Has the statistical analysis been performed appropriately and rigorously?

Reviewer #2: Yes

4. Have the authors made all data underlying the findings in their manuscript fully available (please refer to the Data Availability Statement at the start of the manuscript PDF file)?

Reviewer #2: Yes

5. Is the manuscript presented in an intelligible fashion and written in standard English?

Reviewer #2: Yes

6. Review Comments to the Author

Reviewer #2: You did a nice job responding to the comments.

7. PLOS authors have the option to publish the peer review history of their article (what does this mean?). If published, this will include your full peer review and any attached files.

**Do you want your identity to be public for this peer review?** For information about this choice, including consent withdrawal, please see our Privacy Policy.

Reviewer #2: No
